# Alternative carbon price trajectories can avoid excessive carbon removal

Jessica Strefler [1✉], Elmar Kriegler [1,2], Nico Bauer[1], Gunnar Luderer[1,3], Robert C. Pietzcker [1], Anastasis Giannousakis [1] & Ottmar Edenhofer [1,3,4]

The large majority of climate change mitigation scenarios that hold warming below 2 °C show high deployment of carbon dioxide removal (CDR), resulting in a peak-and-decline behavior in global temperature. This is driven by the assumption of an exponentially increasing carbon price trajectory which is perceived to be economically optimal for meeting a carbon budget. However, this optimality relies on the assumption that a finite carbon budget associated with a temperature target is filled up steadily over time. The availability of net carbon removals invalidates this assumption and therefore a different carbon price trajectory should be chosen. We show how the optimal carbon price path for remaining well below 2 °C limits CDR demand and analyze requirements for constructing alternatives, which may be easier to implement in reality. We show that warming can be held at well below 2 °C at much lower long-term economic effort and lower CDR deployment and therefore lower risks if carbon prices are high enough in the beginning to ensure target compliance, but increase at a lower rate after carbon neutrality has been reached.

[1] Potsdam Institute for Climate Impact Research (PIK), Member of the Leibniz Association, Potsdam, Germany. [2] Universität Potsdam, Potsdam, Germany. [3] Technische Universität Berlin, Berlin, Germany. [4] Mercator Research Institute on Global Commons and Climate Change, Berlin, Germany. ✉email: strefler@pik-potsdam.de

In the Paris Agreement, the United Nations member states have agreed to hold the increase of global mean temperature well below 2 °C and to pursue efforts to limit it to 1.5 °C. Only a tight cumulative budget of admissible carbon dioxide emissions remains to achieve this[1, 2]. Given this tight budget, there is no scenario available that stays below 1.5 °C in 2100 without actively removing carbon dioxide from the atmosphere[3]. Also the majority of 2 °C scenarios rely on large scale deployment of carbon dioxide removal (CDR) technologies (Fig. S1). While CDR may be necessary to achieve the climate target, the scale of CDR usage has been exaggerated in many scenarios. The reason for this is, that many of these scenarios show net carbon removals, i.e. the amount of CDR exceeds the remaining carbon emissions, in the long run. Net carbon removals lead to a decline of global mean temperature, resulting in a peaking of global warming well above the end of century warming level[4] (Fig. S2). If the warming peak were around or above 2 °C, these scenarios would not be consistent with the Paris climate goal of holding warming to well below 2 °C. Therefore, many of the peak-and-decline 2 °C scenarios exhibit a warming peak below 2 °C. The following decline in temperature might be beneficial in terms of climate damages, but comes with the downside of large-scale CDR that is not necessary for target compliance. Challenges and sustainability concerns come with all CDR options[5], and they tend to increase with deployment. In addition, the feasible scale of CDR is uncertain[6] and financing of net-negative emissions may lead to large institutional challenges[7]. Scenarios relying on large amounts of CDR are therefore quite risky.

The recent literature has discussed the problem of massive CDR deployment in stringent climate change mitigation scenarios[8], but focused more on relieving the symptoms rather than tackling the cause in terms of the underlying economics. Obersteiner et al.[9] have called for new scenarios that account for the challenges and uncertainties associated with large-scale CDR deployment and aim to avoid negative side-effects, sketching alternative pathways that would reduce CDR deployment. However, they did not analyze how those pathways could derive from economic and policy considerations. It was shown that additional demand-side policies and assumptions about lifestyle changes[10] or low energy demand[11] reduce the need for bioenergy with carbon capture and storage (BECCS) and benefit sustainability, but the economic drivers for CDR were not directly attacked and carbon removals were partly shifted to afforestation. The endogenous representation of lifestyle changes and energy demand reducing measures in models remains a critical research agenda to better understand the economics of such low demand scenarios. At the same time, lifestyle changes or low energy demand cannot be taken as given. The question remains how the economics of a robust, low-risk mitigation pathway can look like even if lifestyle changes and efficiency measures do not run as deeply as envisaged in these scenarios. To this end, it was shown that a reduction of the discount rate can reduce the temporary exceedance of an end-of-century carbon budget and, consequently, the amount of net-negative emissions[12]. In order to constrain target overshoot directly, it has been proposed to apply carbon budgets only until the time of carbon neutrality instead of the full century[13, 14]. This is in line with the definition of carbon budgets as peak warming budgets in climate research.

Here we go one step further and argue that the combination of finite carbon budgets associated with the temperature limits and the availability of CDR exceeding residual carbon emissions requires revisiting core economics of cost-effective mitigation pathways. The majority of scenarios used in the AR5 assume (explicitly or implicitly) an exponentially increasing carbon price path together with high CDR potential (Fig. S3). This is the main reason for the peak-and-decline temperature trajectories, mirrored by a peak-and-decline in cumulative carbon emissions. The carbon price assumption goes back to the "Hotelling rule"[15]: a price path that rises exponentially with the discount rate is intertemporally optimal for exhausting a finite and exhaustible resource, in our case the finite remaining carbon budget. However, once CDR is introduced to the portfolio of mitigation options, the remaining admissible amount of cumulative gross $CO_2$ emissions is no longer finite, and the Hotelling rule no longer represents an economically optimal solution to stay below a remaining carbon budget at any time. Hence, imposing such a constraint combined with CDR leads to changes in the basic characteristics of the optimal price trajectory.

In cost-benefit-analysis (CBA), the optimal carbon tax looks very different. Golosov et al.[16] find an optimal carbon tax to increase at the rate of GDP growth – which is lower than the interest rate – under the assumption of constant savings and logarithmic utility, if both economic damages increasing linearly with GDP and mitigation costs are taken into account. Starting from the almost immediate warming response to carbon emissions and therefore also the almost immediate and permanent avoidance of damages due to emission reductions, Dietz and Venmans[17] reach the same conclusion. On the contrary, Nordhaus et al.[18] find the optimal carbon tax profile to be almost linear.

In this study we analyze the effects resulting from different shapes of carbon price pathways and the implications for designing economically reasonable climate policies in a cost-effectiveness framework. We investigate the fundamental impact of the shape of the carbon price path on temperature overshoot and CDR deployment. Using the global energy-economy-climate model REMIND[19], we analyze six different scenarios, all assuming current policies until 2020 and a uniform global carbon price thereafter that is adapted to achieve a global cumulative $CO_2$ budget of 1070 Gt $CO_2$[3] from 2018 onwards, which is consistent with a 67% chance of limiting global mean temperature increase to 2 °C (Table 1). First, we consider the economically optimal price path evolving endogenously from the model under the condition that cumulative emissions at no point in time exceed the carbon budget. We then compare different shapes of carbon price trajectories to this benchmark. The closest approximation to this optimal pathway is an exponentially increasing

**Table 1 Scenario definitions.**

| Scenario | Short name | Carbon budget | Rate of carbon price increase |
|---|---|---|---|
| Optimal | OPT | Never exceeded | Endogenous |
| Hotelling to Constant | H2C | Never exceeded | Discount rate until time of net-zero emissions, constant price thereafter |
| Hotelling Overshoot | HOS | Not exceeded in 2100 | Discount rate |
| Hotelling Below | HBL | Never exceeded | Discount rate |
| GDP growth | GDP | Never exceeded | GDP growth rate |
| Linear | LIN | Never exceeded | Such that target is met |

price at the rate of the discount rate until the time of net-zero emissions that is constant thereafter. This is what would be expected as an optimal outcome if there were no path dependencies, i.e. a stable carbon price would lead to stable emissions. As a third and fourth scenario, we consider an exponential price pathway resulting from the Hotelling logic. For this price path, we set two different climate targets with a constraint on peak cumulative vs. end of century cumulative $CO_2$ emissions, reflecting the difference between holding temperature below a given limit throughout the century or returning temperature to this limit by the end of the century. These price pathways are most often found in integrated assessment models (IAMs). Following Golosov[16] and Dietz and Venmans[17], we also consider a carbon price pathway that increases at the rate of GDP growth. Approximately following Nordhaus[18] we consider a linear price path as a simplified alternative that could also be easier to implement politically. Germany for example plans to implement an approximately linear carbon price for all emissions that are not part of the European emissions trading scheme (ETS) from 2021 onwards. The starting level of the linear price path equals the price of the optimal scenarios in 2025 and the annual increase is chosen such that the carbon budget is never exceeded. A linear price path has two free parameters: the starting price and the annual increase. By making the starting price equal to the OPT scenario, we are able to separate effects resulting from the shape of the carbon price path from effects that result from a different level of ambition in near-term climate policy. This construction leads to slightly higher prices than OPT in the first half of the century and lower prices in the second half, but similar levels in 2025 and 2100. It is also very close to the numbers in Nordhaus.

## Results

**Optimal carbon price pathway limits CDR deployment.** Figure 1 shows the resulting total global $CO_2$ emission and temperature trajectories as well as the carbon price and CDR deployment paths until 2100 for all six scenarios (see Fig. S4 for gross $CO_2$ emissions and single option CDR deployment). In a cost-effectiveness framework with an explicit carbon budget limit that is valid at all times, the optimal carbon price pathway shows an exponential increase at the rate of the discount rate until the time when emissions reach net-zero and the carbon budget is exhausted. The price trajectory after this point depends on the available budget and the associated emission and carbon price trajectory before the budget has been exhausted. In our 2 °C scenario, the carbon price first drops and then increases back to a similar level at the end of the century, leading to a stabilization of $CO_2$ emissions around net-zero and therefore only a slight decline of global mean temperature due to declining non-$CO_2$ emissions (Fig. S4). In a simpler model based on marginal abatement cost curves, one would expect a constant carbon price after the budget is exhausted, which would lead to constantly zero emissions. In reality, the energy system might take longer to reach equilibrium. Technologies that are competitive at this carbon price (especially CDR technologies) would continue to be built, leading to further declining emissions and a stabilization at a lower, net-negative level. The exact level of stabilization would depend on the exact pathway before. When comparing the OPT and the H2C scenarios, we see this kind of behavior. In the H2C scenario, CDR deployment continues to increase and emissions continue to decline for another decade after the carbon price has stabilized. However, net-negative emissions are not necessary for target compliance. Therefore, the carbon price in the OPT scenario shows a dip leading to immediate stabilization of emissions. This pattern is even more pronounced for the higher carbon prices in the 1.5 °C pathways (Fig. S5). The H2C, GDP, and LIN scenarios

are all reasonable approximations to the OPT scenario. The range of carbon prices is comparable throughout the century, leading to similar emission pathways, peak temperature, and CDR deployment. They are also similar in terms of economic efficiency (see Fig. S4).The HOS and HBL scenarios on the other hand lead to much higher long-term carbon prices and almost twice the level of CDR deployment. This makes the HBL scenario more costly. The HOS scenario is less expensive, but this comes at the cost of intermittently exceeding the carbon budget and therefore also the 2 °C global mean temperature increase.

However, the approximation of the optimal price pathway does not work for all climate targets. For lower carbon budgets in line with a 1.5 °C target, the emission reductions have to be steeper and the point of carbon neutrality has to be achieved already in 2050 (Fig. S5). This requires much higher carbon prices and a more rapid upscaling of CDR. Consequently, emissions approach the point of carbon neutrality not gradually, but rather at near the quickest possible rate. The optimal carbon price sharply drops after that point, instead of staying almost stable. Such a price path cannot be approximated by any of the other shapes analyzed here. All pathways vary widely in terms of temperature profile, CDR deployment, and economic costs. Only the optimal pathway avoids net-negative emissions and a peak-and-decline of global mean temperature and is able to limit CDR deployment to <10 Gt $CO_2$/yr. All other scenarios eventually reach levels of up to 20 Gt $CO_2$/yr, resulting in significant net-negative emissions and a peak-and-decline shape of global mean temperature. Again the HBL scenarios leads to only half of the cumulative discounted consumption loss of the OPT scenario, but temperatures are higher throughout the century, with a peak difference of 0.2 °C. All other scenarios show costs that are 15–37% higher than OPT.

**Peak temperature determines near-term emission reductions.** We find that the peak temperature limit determines the near-term emission reductions, almost independently of the shape of the carbon price path. Even though the shapes of the carbon price pathways are very different, the absolute value of carbon prices and the resulting emissions trajectories of all scenarios but the HOS are quite similar until 2050. This is consistent with the SR1.5[1], which showed that distinguishing classes of scenarios not only according to end-of-century temperature, but also according to the level of overshoot reduces uncertainty in the necessary 2030 emission reductions. This makes 2030 emission reduction levels necessary for staying well below 2 °C much more robust than previously thought. It is also in line with e.g. Meinshausen et al.[2] who showed that cumulative emissions until 2050 are a robust indicator for the probability of not exceeding 2 °C. To achieve those emission reductions, it is important that the carbon price is sufficiently high already in 2025. The carbon price in 2025 in the OPT scenario is at 36 $/t$CO_2$ 55% higher than in the exponential HOS with 23 $/t$CO_2$ (see Table S3). The higher early carbon prices lead not only to stronger emission reductions and therefore more avoided (below 2 °C) damages, but also to an earlier upscaling of CDR in the period 2030–2050 (Fig. 1b). Total annual CDR deployment increases on average between 2030 and 2050 by 115 Mt $CO_2$ per year in the OPT scenario and reaches ~2.7 Gt $CO_2$/yr in 2050, but only by 60 Mt $CO_2$/yr in the HOS scenario (see Table S4). These scale-up rates are very challenging, but comparable to the current scenario literature[20].

**Hotelling price paths drive long-term CDR and peak-and-decline behavior.** In the second half of the century, the strong carbon price increases of the HOS and HBL scenario lead to 2.4 and 3.8 times the price of the OPT scenario in 2100, respectively. This results in very deep and costly reductions of $CO_2$ emissions

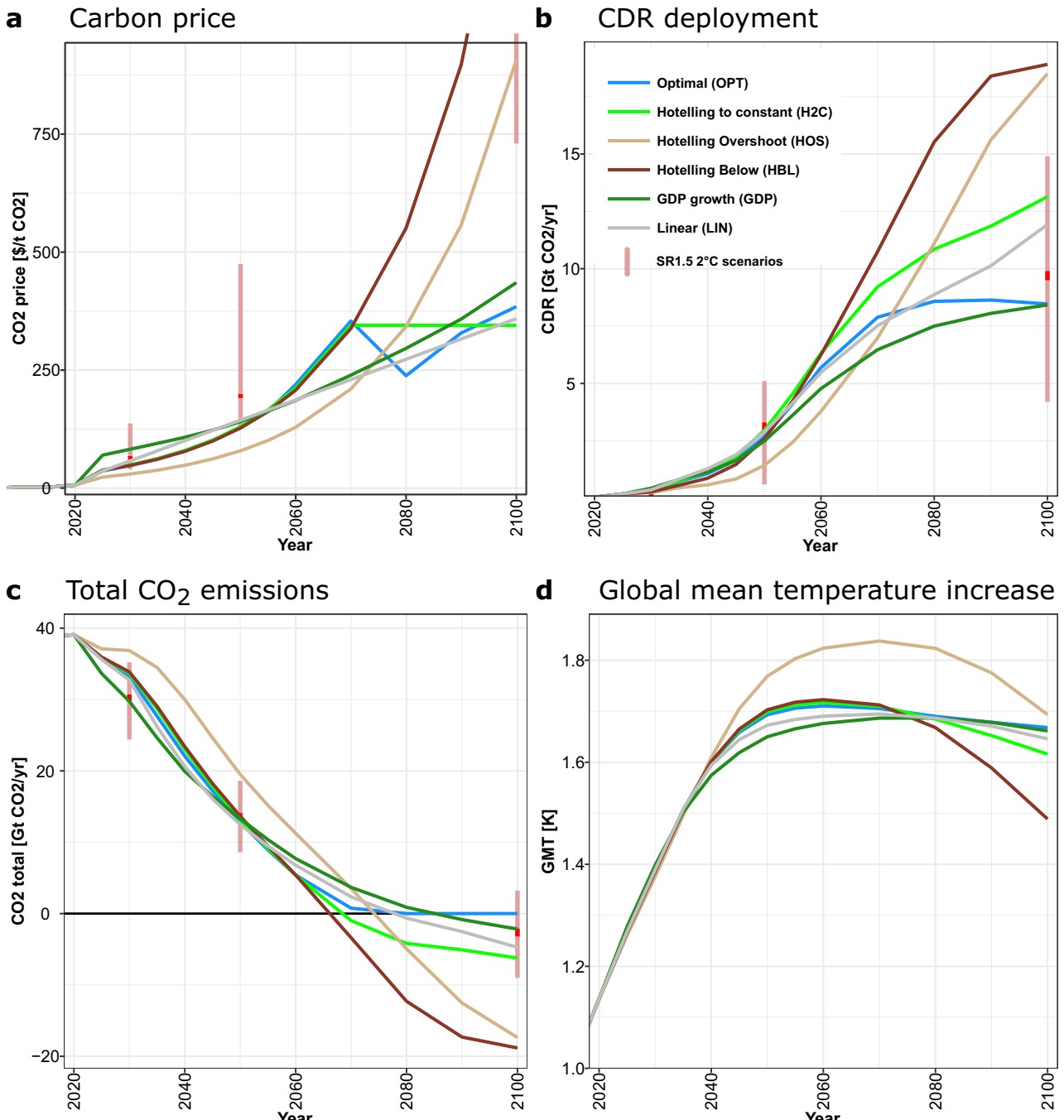

**Fig. 1 Characteristics of the six scenarios. a** Carbon price, **b** Carbon Dioxide Removal deployment, **c** total CO₂ emissions, and **d** global mean temperature increase for the Optimal (blue), Hotelling to Constant (light green), Hotelling Overshoot (light brown), Hotelling Below (dark brown), GDP Growth (dark green), and Linear (gray) scenarios. The red circles and bars show the median and the interquartile range of the SR1.5 2 °C scenarios (i.e. categories "Lower 2C" and "Higher 2C"), respectively.

from fossil fuel burning (see Fig. S4b) and a massive deployment of CDR in the Hotelling scenarios that is twice as high in 2100 than in the OPT scenario. The resulting large net-negative emissions are needed to return temperature to well below 2 °C by the end of the century in case of the HOS scenario, but they would not be needed in the HBL scenario as the temperature peak is already well below 2 °C. The massive scale up of CDR deployment in the second half of the century in both scenarios results from the Hotelling assumption and can be avoided with an only moderate increase of the carbon price after emissions neutrality has been reached.

As compared to the HBL, the OPT achieves the same peak temperature with moderate end-of-century carbon prices and therefore much less CDR and lower long-term economic costs (see Fig. S4h). Cumulative discounted consumption loss from 2020 to 2100 with respect to a scenario of continued current policies is reduced from 1.06% for the HBL scenario to 0.85% for the OPT. The low costs of 0.66% in the HOS scenario come at the expense of exceeding the temperature limit for some time and using large amounts of net-negative emissions to return to it, therefore increasing long-term risks as well as climate impacts.

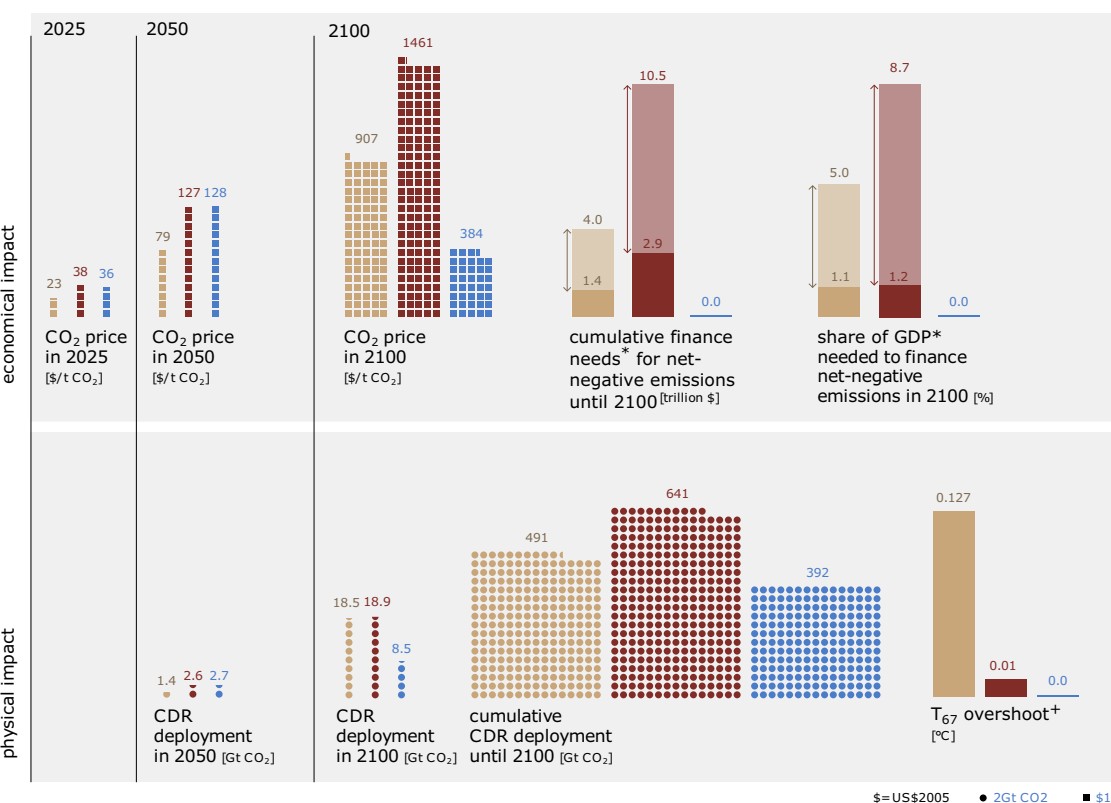

**Fig. 2 Risk profile of Hotelling Overshoot (light brown), Hotelling Below (dark brown), and Optimal (blue) scenarios.** The upper panel shows the economic impact indicators $CO_2$ price (in 2025, 2050, and 2100), finance needs for net-negative emissions in absolute values and as share of GDP. The lower panel shows physical impact indicators Carbon Dioxide Removal (CDR) deployment (in 2050 and 2100), cumulative CDR deployment, and temperature overshoot. The Optimal scenario reduces long-term economic and physical risks and has near- and medium-term risks comparable to the Hotelling Below scenario. *These finance needs for net-negative emissions show upper estimates for the case that CDR is rewarded with the full current carbon price and lower estimates for the case that CDR is rewarded with no more than 250 $/tCO_2, which would avoid high rents for CDR companies. +The temperature overshoot is calculated as the difference of the maximum mean temperature and the mean temperature in 2100 in a scenario where the carbon budget is met in 2100, which is ~1.7 °C.

**Optimal carbon price path reduces long-term risks.** In Fig. 2 we show several short-, medium-, and long-term indicators of economic as well as physical risks. Economic indicators relate to the carbon price and finance needs for net-negative emissions. Physical indicators deal with CDR deployment and the level of temperature overshoot over 2 °C at 67% probability. The combination of all indicators results in a specific risk profile for each of the different scenarios. The OPT and the HBL require a global carbon price of ~36 $/tCO_2 in 2025. This is 55% higher than in the HOS scenario and can therefore increase short-term economic risks. The two scenarios also show similar carbon prices in 2050 that are higher than in the HOS scenario. The higher 2050 carbon prices correspond to a faster transition of the energy system towards a carbon neutral economy and higher deployment of CDR in 2050. In the OPT scenario, however, this does not lead to an increase in the maximum CDR deployment, but rather to an upscaling a decade earlier. This early deployment mainly increases the technical risks of whether CDR technologies can be scaled up in time and at the required pace.

The OPT scenario reduces many of the long-term risks associated with the exponential price pathways. Short- to medium-term risks are similar to those of the HBL scenario. The OPT scenario reduces long-term carbon prices by a factor of 2.4–3.8 and halves maximum annual CDR deployment. The high carbon prices at the end of the century that emerge due to the exponential increase lead to implementation risks. Nation states

may not be willing or able to commit to such high carbon prices. Large CDR deployment is associated with technical, institutional, and sustainability risks. The combination of high carbon prices with large CDR deployment and therefore net-negative $CO_2$ emissions leads to an additional financial risk in the Hotelling scenarios. As long as emissions are net-positive, their costs can be covered by revenues from the residual emissions. As soon as emissions turn net-negative, their costs exceed the incoming carbon revenues. The differential has to be covered either from saved past revenues or from concurrent taxation. If CDR were rewarded at the full current carbon price, the cumulated net present value of finance requirements for net-negative emissions could reach 10.5 trillion US$, with the annual finance needs amounting to 8.7% of gross world product in 2100 in the HBL scenario. However, it seems likely that at very high carbon prices, governments would not reward CDR with the full carbon price but try to reduce costs by using an auctioning system. Instead of integrating CDR into an emissions trade scheme, the desired amount of CDR could be offered for bidding with the cheapest offers getting the bid. Such a system would aim to reward CDR with prices that are closer to the actual costs of CDR instead of prices that reflect the marginal abatement costs of emissions and thus reduce the rents of CDR suppliers[7]. If such a system would work and CDR would be awarded at e.g. no more than a mean price of 250 $/tCO_2, the cumulated net present value of these finance needs discounted to 2020 values would be reduced to <3

trillion US$ or 1.2% of gross world product in 2100 in the HBL scenario. The differences in the finance needs show that a maximum carbon removal bonus paid to CDR companies is a crucial feature of future long-term climate policies, particularly in case of an overall policy framework that would lead to very high carbon prices. These finance requirements are a critical barrier to the institutional feasibility of net-negative $CO_2$ emissions. Figure 2 shows that such finance requirements are much larger in the HOS scenario, which relies on massive CDR deployment at the end of the century for meeting the well below 2 °C target at least in 2100. If the anticipated high net-negative emissions could then not be realized in the end, the target would remain breached. Finally, climate damages are not only related to end-of-century warming, but also to the full temperature pathway including peak warming. Overshooting the temperature limit could increase the risk to trigger tipping points, and the higher temperature level that is sustained for decades could lead to increased climate impacts.

## Discussion
The inclusion of large-scale CDR availability in climate change mitigation scenarios has invalidated the Hotelling assumption for calculating cost-optimal pathways. Yet this price path is still used in many climate change mitigation scenarios and leads to high end-of-century carbon prices and an exaggerated CDR demand. Remaining below 2 °C requires fast and deep emission reductions, and therefore a high immediate carbon price. Once emission neutrality has been reached, a further increase in mitigation ambition is not necessary and is in fact counterproductive as it leads to large-scale CDR deployment and the associated substantial risks. If high ambition for the current and next decades is complemented by a more moderate increase in the second half of the century, the climate target can be maintained with limited risks. The precise shape of the carbon price pathway is of lesser importance. All alternative carbon price pathways analyzed in this study show similar results in terms of timing of early emission reduction, CDR deployment, and peak temperature and also similar consumption losses.

The use of exponential $CO_2$ price paths in climate change mitigation scenarios has led to a preponderance of scenarios with a peak-and-decline trajectory enabled by the combination of high $CO_2$ prices with heavy use of CDR. As the available range of knowledge about mitigation pathways influences or even determines the range of political choices, it is important to assess also alternative scenarios with less CDR reliance and lower long-term risks. In this study, we have investigated such alternatives with long-term carbon prices much lower than in the range of scenarios assessed in the AR5. This reduces technical, social, and ecological risks due to reliance on large-scale CDR as well as governance and finance risks related to the high carbon price itself and the finance needs of high net-negative emissions.

Based on our insights, choosing a different price path than Hotelling is possible and seems wise for the reasons of avoiding temperature overshoot, avoidance of massive CDR, and reduced climate change risks. This is different to the analysis by Emmerling et al.[12] that only studied variations of the growth rate of carbon prices and the impact on carbon budget overshoot, but did not question the shape of the price trajectory. Our results indicate that the change in the shape is a crucial degree of freedom to approximate the optimal carbon price path. The approximation is particularly better in the near term.

To limit maximum long-term temperature to well below 2 °C, the near-term carbon price needs to be sufficiently high, and substantial CDR requirements (~4 Gt $CO_2$ in 2050, >10 Gt $CO_2$ in 2100) remain. This is consistent with Strefler et al.[8] who showed that economic costs start to increase significantly if <5 Gt $CO_2$/yr CDR are available.

In this study, we have shown why under the inclusion of CDR a Hotelling price path no longer represents the economically optimal solution and we have laid out some general principles to avoid large-scale CDR and reduce long-term risks. The optimal carbon price path derived in this study is only cost-optimal to achieve the 2 °C target. It still ignores (i) damages below 2 °C and (ii) co-benefits from climate policy. The presence of non-negligible impacts below the climate target[21, 22] or the consideration of risk and uncertainty[23] could justify higher near-term carbon prices[17]. Benefits from emission reductions e.g. for air pollution could also lead to higher carbon prices than those derived from climate policy[19].

We conclude that a carbon price path that starts high and rises only moderately after emission neutrality has been reached allows to stay well below 2 °C without massive CDR deployment and provides robust projections of emissions pathways across varying perceptions of the timing of well below 2 °C. This reduces economic as well as technological and institutional risks.

## Methods
**Study design**. We use the global multi-regional energy-economy-climate model REMIND[19, 24, 25] Version 2.1.0 for our analysis. REMIND is open source and available on GitHub at https://github.com/remindmodel/remind. The technical documentation of the equation structure can be found at https://rse.pik-potsdam.de/doc/remind/2.1.0/. In REMIND, each single region is modeled as a hybrid energy-economy system and is able to interact with the other regions by means of trade. Tradable goods are the exhaustible primary energy carriers coal, oil, gas and uranium, a composite good, and emission permits.

The economy sector is modeled by a Ramsey-type growth model which maximizes utility, a function of consumption. Labor, capital, and end-use energy generate the macroeconomic output, i.e. GDP. The produced GDP covers the costs of the energy system, the macroeconomic investments, the export of a composite good, and consumption.

The energy sector is described with high technological detail. It uses exhaustible and renewable primary energy carriers and converts them to final energies as electricity, heat, and fuels. Various conversion technologies are available, including technologies with carbon capture and storage (CCS). Regional annual CCS deployment is limited to 0.5% of total storage capacity. This limits total global CCS use to ~20 Gt CO2/yr.

The Hotelling price pathways increase at 5% per year and the starting value in 2025 is adapted iteratively such that the carbon budget is reached. This is nearly equivalent to the explicit formulation of a constraint on the 2018–2100 CO2 budget. The linear price pathway starts at the level of the exponential below scenario in 2025 and the annual increase is adapted iteratively such that the carbon budget is reached. For the exponential price path, we set two different climate targets with a constraint on peak cumulative vs. end of century cumulative $CO_2$ emissions, reflecting the difference between holding temperature below a given limit throughout the century or returning temperature to this limit by the end of the century. The resulting temperatures are calculated using MAGICC6[26].

**CDR technologies**. In addition to CCS with fossil fuels and in the industry sector, three CDR options are available: afforestation and reforestation[27], bioenergy with CCS[28, 29] (BECCS), and direct air capture with CCS[30] (DACCS).

$CO_2$ emissions from afforestation and reforestation are derived from the land-use optimization model MAgPIE4.0[31, 32] (Model of Agricultural Production and its Impact on the Environment). MAgPIE is a spatially explicit, global land-use allocation model and projects land-use dynamics in 10-year time steps until 2095 using recursive dynamic optimization. Land-based mitigation in MAgPIE is incentivized by an exogenously given tax on GHG emissions. The tax is consistent with a SSP2-2.6 scenario[33]. While the GHG price renders deforestation and the conversion of pasture to cropland more costly, $CO_2$ removal through afforestation is rewarded and lowers the costs in the objective function of the MAgPIE model. The trade-off between land expansion and yield increases is treated endogenously in the model. To derive the potential and costs for afforestation and reforestation, a bioenergy demand consistent with a SSP2-2.6 scenario was assumed so that the land competition between afforestation and bioenergy is taken into account. Only 50% of the carbon price seen in the energy system is used to derive the $CO_2$ emissions from afforestation and reforestation to account for issues of permanence, i.e. the possibility that the sequestered $CO_2$ is released again due to forest fires or inadequate forest protection.

BECCS is the CDR technology most widely used in the AR5 scenarios and the only CDR technology that provides sizeable energy instead of consuming it. The idea of BECCS is to turn biomass grown on land carbon-negative by capturing the

emissions arising during combustion or the refinery process. MAgPIE captures $CO_2$ emissions from land conversion (e.g. from forest) into biomass plantations including soil carbon emissions, and $N_2O$ emissions from fertilizer use, but does not account for emissions from soil degradation. BECCS can be used for electricity, hydrogen, gas, or liquid fuel production with different carbon capture rates (see SI). Bioenergy supply curves are derived from the model MAgPIE. In REMIND, an additional tax on bioenergy of 100% of the bioenergy price is imposed to account for sustainability issues not included in the model, e.g. biodiversity loss or water consumption.

DACCS captures $CO_2$ directly from the ambient air. We rely on the literature review performed in Broehm et al.[30] for techno-economic parameterization. We assume a demand of 10 GJ/t $CO_2$ heat and 2 GJ/t $CO_2$ electricity. In the model, natural gas or $H_2$ can be used to generate the required heat. If natural gas is used, the resulting $CO_2$ emissions are assumed to be captured with a capture rate of 90%. An estimated 100 $/ $tCO_2$ investment costs (excluding energy costs and costs for carbon storage) makes it a rather expensive option compared to both BECCS and afforestation, but on the upside DACCS is less dependent on the location and requires only little land.

## Data availability
The datasets generated and analyzed in this study and the plot routines for creating the figures are available at Zenodo repository https://doi.org/10.5281/zenodo.3999986.

## Code availability
REMIND is open source and available on GitHub. The model version used in this study is 2.1.0, which can be downloaded https://github.com/remindmodel/remind/releases/tag/v2.1.0.

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

## Acknowledgements
The research leading to these results has received funding from the German Research Foundation (DFG) Priority Programme (SPP) 1689 (CEMICS2) and from the European Union's Horizon 2020 research and innovation programme under grant agreement No 821124 (NAVIGATE). We thank Bernd Grether for designing Fig. 2.

## Author contributions
J.S., E.K., N.B., and O.E. designed the concept of the paper. J.S., E.K., G.L., and R.C.P. designed the scenarios. J.S. and A.G. produced the scenarios. J.S. analyzed the scenarios, prepared the figures, and wrote the manuscript, with input from all authors.

## Funding

## Competing interests
The authors declare no competing interests.
