## [Peer Review File · Nature Communications]

Reviewers' comments:

Reviewer #1 (Remarks to the Author):

The manuscript "Alternative carbon price trajectories can avoid excessive carbon removal" argues that using different intertemporal carbon price trajectories leads to very different mitigation profiles and policy mixes, including much lower CDR use in the long run, potentially. It does so by applying an IAM model (REMIND) and compares exponential carbon price increases vs. a linear carbon tax over time, in peak budget or century full carbon budget approaches.

The manuscript is a short and well-written piece on an important topic. It speaks to the IAM literature and CDR, and the economics of optimal carbon pricing, where recently Daniel et al. (2019) added to the discussion about increasing, exponential, constant or even decreasing optimal carbon taxes. As such it has some merit, and the execution seems very well done.

In terms of innovation, however, I am less confident on the value of the paper: notably, it (for most of the paper) compares an exponential with a linear increasing carbon tax. It would have been interesting to see a constant carbon tax, or maybe even decreasing rates. As it is, "alternative" here only means exponential vs. linear and thus is not very general, even though not unrealistic I admit.

Another issue I have with the paper is potential overselling: About your "Two assumptions" in the abstract: not a finite budget and not considering economic damages below the target: they are probably correct in my opinion, and from reading the abstract it is a very interesting paper. However, after reading the paper, you don't actually address these points very much: For the Hotelling not holding case, an analytical model would be interesting so see what optimal path comes out instead. (You claim on l.69 that the budget is "no longer finite", but I don't think it is a problem of finiteness here (agreed it can theoretically be very large, but the problem is more the boundedness of (in the case of Hotelling) extraction by zero from below. Would be actually interesting to see if it can be solved analytically. As for your second assumption, economic damages: agreed they could change the picture, but as I understand the model, damages are NOT included in this paper. So both main claims are not really addressed by the exercise. This is another major shortcoming.

Yet overall, it is a nice written piece and hence I can recommend a revision.

Suggestions to improve the manuscript:

- The figures are nice, but the quality is very bad please improve to make it readable.
- It would be good to have four scenarios, even if the two linear ones are similar for clarity. It is not easy to get always only from colors the differences.
- Wording "peak and decline" never read this term before, why did you not chose overshoot for simplicity (OK to leave just curious)
- Scenario 3 reads a bit weird: Why you chose the same initial price? Why not considering alternative starting values and hence increments? Or even different growth rates? At least for an SI
- L71: your maximum assumption of CCS to 0.5% per total reservoir, is there a basis or calibration for that? As it could be very limiting at that value you give (20 GtCO₂)
- Table 52: could you compare with like e.g., the UK report on DAC or others for reference and comparison? Seems could be on the lower end the costs.

Daniel, Kent D., Robert B. Litterman, and Gernot Wagner. "Declining CO2 Price Paths." Proceedings of the National Academy of Sciences, October 1, 2019, 201817444.
<https://doi.org/10.1073/pnas.1817444116>.

Reviewer #2 (Remarks to the Author):

Review report: Strefler et al. Alternative carbon price trajectories can avoid excessive carbon removal.

The paper presents a carbon price (tax?) experiments with three distinct scenarios. The advantages of the linear price development schedule are presented. It then presents a 2 stage decision tree framework to determine carbon prices and climate mitigation targets.

General remarks

The paper presents two weakly connected pieces. One being the exogenous carbon price experiment with the REMIND model and the other the decision tree framework.

Let me first comment on the "exogenous" carbon price experiment:

Driving climate abatement scenarios with exogenously determined linear, exponential, hyperbolic etc... carbon prices is common practice in quantitative assessment modeling of climate mitigation. The most pertinent examples is the GLOCAF model operated by the former UK Office of Climate Change, but there is a plethora of other models that mostly operate with Marginal Abatement cost curves (or dynamically linked schedules) that conduct such experiments. This is a practice which to my knowledge is at least 25 years old. The general conclusions from this paper are of course the same here and there.

What makes this study distinct from the former experiments with linear and exponentially increasing carbon prices is that they treat costs (technological change) endogenous to the model and provide macro-economic feedbacks in one model rather than through a soft linkage. I would consider this technical difference relatively small given the rather coarse technology and regional resolution of the REMIND model in terms of deriving climate mitigation costs. Ex-ante I would not expect fundamental differences in relevant insights compared to a MAC curve – CGE linkage.

I am somehow puzzled that the paper still uses the Hotelling rule in 2 out of three scenarios although

the authors rightly argue that Hotelling is an inappropriate methodology to assess the use of a carbon budget. The Hotelling rule was designed to provide insights on how to optimally extract a non-renewable resource. In the case of climate mitigation, the static carbon budget is considered to stand for the non-renewable resource. According to my understanding the Hotelling rule would only apply if no negative emission technologies were allowed. When you extract a mine you will not throw ore back into the mining pit – would you!? This is similar to making up for initial over extraction. I would very much welcome a very clear treatment of this issue in this paper. Clearly, the application of Hotelling has lost its application. Of course, it is still fine to calculate whatever scenario based on Hotelling, but then I would like to see a clear justification for why this rule is of much interest even as a benchmark. Maybe the best justification is that it has been used by the majority of models tasked with the problem of pathways for 1.5 and 2 C thus far. However, the paper would benefit from a deeper treatment of the economic and ethical meaning of such calculations and the literature it produced.

I was also somewhat surprised that the REMIND model works with a flat 5% discount rate or any other time consistent and constant discount rate. I always thought that the economics of REMIND is closer to other more economic models (in contrast to the technology-rich techno-cost economic models) and use the Ramsey formula for discounting. A discussion on Ramsey and Hotelling to determine the discounting methodology and thereby the rate of carbon price development would be useful and a clear motivation should be given why a Hotelling rule (if the authors still insist on the use of it) is the best choice to conduct the benchmarking exercise with the linear carbon price. It would be more interesting to conduct a much richer experiment with varying the discount rate (although it was done already!) all the way to zero or even let it go negative. Negative and time-varying discount rates have been proposed in the more theoretical literature and simpler IAMs. Why is it of interest to look at a linear carbon price only when we analyze a highly dynamic and probably rather non-linear system??

Furthermore, it is not clear why the use of a Ramsey type model is not conducive to a calculation of some sort of social cost of carbon and why a SCC calculation was not used instead of the Hotelling rule. In short, the authors should not only present a pure description of three carbon price trajectories, but actually give a strong and convincing justification and explanation why in the context of the REMIND model and the climate change problematique in general the three scenarios are providing novel insights. Is there a convincing grounding for a linear price pathway in economic theory or by first principles?

Currently, the paper reads more like a cookbook recipe where one finds three-carbon price trajectory that fulfills a set of criteria and then stick those into an existing model (with minimal changes) and present the results. In fact, the basic message could have been developed much more transparently with a much simpler model.

It appears that the success of near term carbon prices impacts significantly the long-run results. However, from the paper and from the documentation of the REMIND model it is difficult to replicate the near-term abatement costs or at least a good proxy of it. It would be a great service to the readers

to see a comparison of for example levelized costs of electricity production implicitly assumed in the model with those that are actually observed in reality today (see e.g. this FORBES article pointing to industry analysis)

Likewise, the impact of technological learning rate assumptions on the modeled results is hardly mentioned. The paper would really benefit from a more analytical treatment of the problem presented. Technological learning would also provide easy linkage to the 2 stage decision framework later in the paper. Clearly, the shape of MACs, learning rates and discount rates and the resulting carbon prices interact dynamically calling for a simple analytical model to present the problem before entering exogenous carbon prices into a complicated model.

I did not dig deep enough into the REMIND model and I do not expect the average reader of a NatureComms article to do so. Therefore, it would be interesting to present, in addition, to the consumption loss a few macro-economic indicators such as interest rate, CPI, the share of government expenditures (see minor comment below), the share of industrial production to better comprehend the 3 scenario situations. The main issue I am after here is that as a reader of the paper I get seemingly convinced (mainly because of a clever choice of indicators) that a suboptimal carbon price trajectory is the best choice. The paper should elaborate explicitly and quantitatively on the issue of economic suboptimality and trade-offs of the linear price path e.g. welfare loss.

The decision process presented in the discussion section seems rather decoupled and it seems a little counterintuitive that "The two decisions of immediate carbon pricing and the price increase after 2050 are independent decisions". Technically, this calls for a discontinuity of the carbon price in 2050. There are decision proposals already in the literature mostly stemming from a 2stage stochastic optimization calculus coming to rather different and more elaborate conclusions. I would strongly recommend that the authors familiarize themselves with this literature and embed their proposal in a setting of 2 stage decision making. Almost surely this will necessitate the authors to propose their own 2 stage decision model which then will be more directly connected to the REMIND model. I am really struggling to see the connection with the numerical results presented above and I am tempted to recommend dropping the discussion section entirely. In fact, I recommend that the 2 stage decision model should be developed with more care and thought for a separate paper.

Minor issues:

If I evaluate the expenditures necessary to finance 20 GtCO₂ in 2100 (S4) at a price of 1500 (Fig2) I get 30 Trillion dollars which make ~35% of today's Global Economic Output. How, realistic is such a scenario even if global GDP will be 5-10 fold in 2100?

The figures in the main text are of poor quality. In some cases, the units on the scales are barely readable.

Figure S3 should not be plotted on a logarithmic scale to ensure better and direct comparability with

Fig1

There are still some typos in the text.

Figure captions in the SI are too short and lack explanation

Reviewer #1 (Remarks to the Author):

The manuscript “Alternative carbon price trajectories can avoid excessive carbon removal” argues that using different intertemporal carbon price trajectories leads to very different mitigation profiles and policy mixes, including much lower CDR use in the long run, potentially. It does so by applying an IAM model (REMIND) and compares exponential carbon price increases vs. a linear carbon tax over time, in peak budget or century full carbon budget approaches.

The manuscript is a short and well-written piece on an important topic. It speaks to the IAM literature and CDR, and the economics of optimal carbon pricing, where recently Daniel et al. (2019) added to the discussion about increasing, exponential, constant or even decreasing optimal carbon taxes. As such it has some merit, and the execution seems very well done.

Thank you for this evaluation.

In terms of innovation, however, I am less confident on the value of the paper: notably, it (for most of the paper) compares an exponential with a linear increasing carbon tax. It would have been interesting to see a constant carbon tax, or maybe even decreasing rates. As it is, “alternative” here only means exponential vs. linear and thus is not very general, even though not unrealistic I admit.

In response to this comment and also others below, we have added a cost-optimal carbon tax pathway that is calculated endogenously in the model to limit cumulative CO2 emission below the budget implied by the 2°C target. In addition, we have added a scenario where the carbon tax increases at the rate of GDP growth, as suggested e.g. in Golosov et al. (2014) and Dietz and Venmans (2019). The constant or even decreasing rates in Daniel et al. are due to their formulation of utility that includes uncertainty and the consideration of damages. In our model, we consider neither climate damages nor uncertainty explicitly, but work with a cost-effectiveness framework. In such a framework, constant carbon prices lead to favorable climatic outcomes (i.e. a global mean temperature that increases slower than in other scenarios), but the cumulative discounted consumption loss is by a factor 2.5 higher than the most expensive Hotelling Below scenario. This does not seem a viable pathway for IAMs, and to not increase the number of scenarios further we prefer to not include a constant carbon price. Considering climate damages and uncertainty could indeed lead to higher near-term carbon prices, and we have included this point in the discussion.

Another issue I have with the paper is potential overselling: About your “Two assumptions” in the abstract: not a finite budget and not considering economic damages below the target: they are probably correct in my opinion, and from reading the abstract it is a very interesting paper. However, after reading the paper, you don’t actually address these points very much: For the Hotelling not holding case, an analytical model would be interesting so see what optimal path comes out instead. (You claim on l.69 that the budget is “no longer finite”, but I don’t think it is a problem of finiteness here (agreed it can theoretically be very large, but the problem is more the boundedness of (in the case of Hotelling) extraction by zero from below. Would be actually interesting to see if it can be solved analytically.

As indicated above, we derived a cost-optimal carbon pricing pathway in presence of a not-to-exceed carbon budget. We agree with the reviewer that it would be valuable to complement this with an analytical model. However, in our case we are mainly interested in the impact of the carbon price pathway on CDR deployment and whether a less risky pathway can be chosen that approximates the efficient solution regarding both emissions and economic impacts. For this project, features such as different CDR options at different costs (which may increase with deployment, e.g. in case of BECCS where the supply of more bioenergy comes at increasing costs) and with different limitations are important. Path dependencies and a somewhat realistic, slow scale-up of CDR are crucial. These aspects cannot be represented in an analytical model based on marginal abatement cost curves. This would be a full new project that would certainly be interesting to pursue, but that we would consider outside the scope of this paper.

As for your second assumption, economic damages: agreed they could change the picture, but as I understand the model, damages are NOT included in this paper. So both main claims are not really addressed by the exercise. This is another major shortcoming.

We see the point of the reviewer that this might be misleading and removed it from the abstract. We now only discuss the possible impacts of considering climate damages in the discussion part of the paper.

Yet overall, it is a nice written piece and hence I can recommend a revision.

Suggestions to improve the manuscript:

- The figures are nice, but the quality is very bad please improve to make it readable.

We improved the figure quality

- It would be good to have four scenarios, even if the two linear ones are similar for clarity. It is not easy to get always only from colors the differences.

We have now added another three scenarios, all of them stabilization scenarios. The linear case is now only one out of four alternatives, so this probably does not apply anymore in the new version.

- Wording “peak and decline” never read this term before, why did you not chose overshoot for simplicity (OK to leave just curious)

“Peak and decline” refers to the shape of the temperature curve. Such a curve does not necessarily have to overshoot the target, the peak may be below e.g. the 2°C limit. In that case, the shape would still be peak and decline, but there would be no overshoot over the target.

- Scenario 3 reads a bit weird: Why you chose the same initial price? Why not considering alternative starting values and hence increments? Or even different growth rates? At least for an SI

The linear scenario has two independent parameters, the start value and the growth rate. We chose to adapt the growth rate to reach the target, therefore the starting value can be arbitrarily chosen. In an extreme case, one would arrive at constant or even decreasing carbon prices. Here we wanted to analyze the impact of the shape of the carbon price trajectory, and not the impact of the starting value. As the other, new scenarios also start at a similar level, we found this choice to make the linear scenario more easily comparable to the other scenarios.

- L71: your maximum assumption of CCS to 0.5% per total reservoir, is there a basis or calibration for that? As it could be very limiting at that value you give (20 GtCO₂)

There are limits to the possible injection pressure, which imply also a limited injection rate. It is quite likely that this injection rate is highly site-specific. Given the increasing risk for seismic activity for higher injection pressures, a lower injection rate can be interpreted as limiting these risks. Our specific assumption is an expert opinion by Dr. Gerling, who was many years the chair of the German Federal Institute for Geosciences and Resources (BGR). This figure was documented in an interview that was available on a website (<http://www.iz-klima.de/aktuelles/archiv/news-2010/mai/news-05052010-2/>). We checked the interview’s availability, but it has been removed from the website. We sent an enquiry to the host, but that was not answered for several weeks. The scientific discussion about the injection rate is ongoing, but there is – as far as we can see – no convergence of the geo-scientific research community towards a best-guess value or a range.

As for our study, the limit on the injection rate is only binding in the last years of the Hotelling Below (HBL) scenario (see Fig. of total CCS use below). It has therefore hardly any impact on the results of our study, both qualitative and quantitative.

- Table 52: could you compare with like e.g., the UK report on DAC or others for reference and comparison? Seems could be on the lower end the costs.

We use the numbers that Climeworks published as their assumption once DAC is produced in series. Even at this price, DAC is mainly used in the Hotelling scenarios. Sensitivities for the Hotelling Below (HBL) scenario, i.e. the scenario with the most usage of DAC, with two and four times the capital costs of DAC but the same carbon price show that if DAC gets more expensive, it is to some extent replaced by BECCS, with little impact on total emissions or economic costs.

Daniel, Kent D., Robert B. Litterman, and Gernot Wagner. "Declining CO2 Price Paths." Proceedings of the National Academy of Sciences, October 1, 2019, 201817444. <https://doi.org/10.1073/pnas.1817444116>.

Reviewer #2 (Remarks to the Author):

Review report: Strefler et al. Alternative carbon price trajectories can avoid excessive carbon removal.

The paper presents a carbon price (tax?) experiments with three distinct scenarios. The advantages of the linear price development schedule are presented. It then presents a 2 stage decision tree framework to determine carbon prices and climate mitigation targets.

General remarks

The paper presents two weakly connected pieces. One being the exogenous carbon price experiment with the REMIND model and the other the decision tree framework.

Let me first comment on the “exogenous” carbon price experiment:

Driving climate abatement scenarios with exogenously determined linear, exponential, hyperbolic etc... carbon prices is common practice in quantitative assessment modeling of climate mitigation. The most pertinent examples is the GLOCAF model operated by the former UK Office of Climate Change, but there is a plethora of other models that mostly operate with Marginal Abatement cost curves (or dynamically linked schedules) that conduct such experiments. This is a practice which to my knowledge is at least 25 years old. The general conclusions from this paper are of course the same here and there.

What makes this study distinct from the former experiments with linear and exponentially increasing carbon prices is that they treat costs (technological change) endogenous to the model and provide macro-economic feedbacks in one model rather than through a soft linkage. I would consider this technical difference relatively small given the rather coarse technology and regional resolution of the REMIND model in terms of deriving climate mitigation costs. Ex-ante I would not expect fundamental differences in relevant insights compared to a MAC curve – CGE linkage.

We agree with the reviewer that it is preferable to have a model as simple as possible. However, in our case we are mainly interested in the impact of the carbon price pathway on CDR deployment and whether a less risky pathway can be chosen that approximates the efficient solution regarding both emissions and economic impacts. For this project, features such as different CDR options at different costs (which may increase with deployment, e.g. in case of BECCS where the supply of more bioenergy comes at increasing costs) and with different limitations are important. This is usually not explicitly covered in MAC curves, which treat mitigation options separately and therefore cannot consider the interactions, trade-offs, and potential synergies. Also the double role of BECCS, which supplies low-carbon energy and at the same time serves as a CDR option is hard to represent using MAC curves. Also, in the context of different time profiles of carbon prices path dependencies need to be considered for the deployment of CDR as time for upscaling the technologies needs to be factored in. REMIND is an intertemporal general equilibrium model with a very high technological resolution of more than 80 energy conversion technologies. The evolving optimal carbon price shows a decline after the budget is exhausted. This is due to the path dependencies in the model and would not be expected in a simple MAC curve – CGE linkage.

I am somehow puzzled that the paper still uses the Hotelling rule in 2 out of three scenarios although the authors rightly argue that Hotelling is an inappropriate methodology to assess the use of a

carbon budget. The Hotelling rule was designed to provide insights on how to optimally extract a non-renewable resource. In the case of climate mitigation, the static carbon budget is considered to stand for the non-renewable resource. According to my understanding the Hotelling rule would only apply if no negative emission technologies were allowed. When you extract a mine you will not throw ore back into the mining pit – would you!? This is similar to making up for initial over extraction. I would very much welcome a very clear treatment of this issue in this paper. Clearly, the application of Hotelling has lost its application. Of course, it is still fine to calculate whatever scenario based on Hotelling, but then I would like to see a clear justification for why this rule is of much interest even as a benchmark. Maybe the best justification is that it has been used by the majority of models tasked with the problem of pathways for 1.5 and 2 C thus far. However, the paper would benefit from a deeper treatment of the economic and ethical meaning of such calculations and the literature it produced.

That is exactly the point of this paper. Hotelling has been used by the majority of model based scenarios used to generate pathways for 1.5°C and 2°C allowing for overshoot flexibility thus far, even though it is not appropriate (anymore). We fully agree with the reviewer that the Hotelling rule is valid for scenarios either without negative emission technologies or for scenarios where an overshoot over the temperature or carbon budget limit is allowed. However, now that CDR is used in the majority of scenarios, the Hotelling rule for the carbon price path cannot be used anymore, if the overshoot is to be avoided. The resource economics literature never discussed this issue for good reason, because it is not possible to extract more of a physical resource than what is actually available. The carbon budget in combination with CDR goes beyond the original Hotelling problem and the overshoot flexibility is not necessarily desirable due to the additional climate risks. We use this as a starting point precisely because it has been used in the majority of scenarios. We have revisited the manuscript on that point and carefully revised the text to better communicate the motivation for our study, e.g. in the introduction we state: “The majority of scenarios used in the AR5 assume (explicitly or implicitly) an exponentially increasing carbon price path together with high CDR potential (Fig. S3). This is the main reason for the peak-and-decline temperature trajectories, mirrored by a peak-and-decline in cumulative carbon emissions.”

I was also somewhat surprised that the REMIND model works with a flat 5% discount rate or any other time consistent and constant discount rate. I always thought that the economics of REMIND is closer to other more economic models (in contrast to the technology-rich techno-cost economic models) and use the Ramsey formula for discounting. A discussion on Ramsey and Hotelling to determine the discounting methodology and thereby the rate of carbon price development would be useful and a clear motivation should be given why a Hotelling rule (if the authors still insist on the use of it) is the best choice to conduct the benchmarking exercise with the linear carbon price. It would be more interesting to conduct a much richer experiment with varying the discount rate (although it was done already!) all the way to zero or even let it go negative. Negative and time-varying discount rates have been proposed in the more theoretical literature and simpler IAMs. Why is it of interest to look at a linear carbon price only when we analyze a highly dynamic and probably rather non-linear system??

We thank the reviewer for this useful comment. The reviewer is absolutely right here, REMIND does use the Ramsey formula for discounting via the model endogenous interest rate. The REMIND model can be used to impose a carbon budget with full overshoot flexibility, which implies that the resulting carbon price increases with the interest rate as is suggested by the Hotelling model. In this study, we implemented carbon price paths with certain specifications such as the 5% annual growth rate. This is close to the internal discount rate and matches what is used by the majority of IAMs. In the present study the only direct implementation of a carbon budget is the case

“optimal”, in which the policy framework requires to meet the carbon budget without overshoot. In this revised version, we also analyze a scenario where the carbon price increases at the GDP growth rate. For the purpose of the present study it is more appropriate to vary the time profile of the carbon price trajectory than to vary parameters that affect the discounting of the overall economy, because this leads to changes in the underlying economy and the long-term growth path. The main goal of this paper is to test alternative carbon price pathways that right-size the need for short-term ambition, the timing of CDR development and the long-term CDR requirements as we feel that these are urgently needed.

Furthermore, it is not clear why the use of a Ramsey type model is not conducive to a calculation of some sort of social cost of carbon and why a SCC calculation was not used instead of the Hotelling rule. In short, the authors should not only present a pure description of three carbon price trajectories, but actually give a strong and convincing justification and explanation why in the context of the REMIND model and the climate change problematique in general the three scenarios are providing novel insights. Is there a convincing grounding for a linear price pathway in economic theory or by first principles?

In this revision, we have added an economically optimal price pathway for the case without carbon budget overshoot and place less weight on the linear pathway. We have added a section in the introduction where we discuss different carbon tax pathways from the literature. An approximately linear price pathway was suggested to be optimal by Nordhaus et al. (2010) as a result of a cost-benefit analysis. Golosov et al. (2014) and Dietz and Venmans (2019) have suggested a carbon price increase that equals the GDP growth rate and is generally lower than the interest rate, which we have included in an additional scenario. Also, the linear price path is a very simple alternative that may thus be attractive to policy makers and is in fact discussed in the German Klimaschutzplan. Despite its simplicity, it approximates the optimal pathway surprisingly well, at least for a well below 2°C target.

Currently, the paper reads more like a cookbook recipe where one finds three-carbon price trajectory that fulfills a set of criteria and then stick those into an existing model (with minimal changes) and present the results. In fact, the basic message could have been developed much more transparently with a much simpler model.

The condition of staying below the carbon budget at all times in combination with the availability of CDR leads to a discontinuity in the carbon price growth rate. Such a discontinuity would be difficult to treat analytically. One of the main focal points in our paper is to show that the CDR deployment and the resulting net-negative emissions have been exaggerated in many of the existing 2°C scenarios. We show that this is due to false assumptions of the underlying carbon price, which is still widely used. We currently do not see how the impact of the carbon price trajectory on CDR deployment could have been derived with a simpler model as many interdependencies, limitations, and path dependencies have to be taken into account.

It appears that the success of near term carbon prices impacts significantly the long-run results. However, from the paper and from the documentation of the REMIND model it is difficult to replicate the near-term abatement costs or at least a good proxy of it. It would be a great service to the readers to see a comparison of for example levelized costs of electricity production implicitly assumed in the model with those that are actually observed in reality today (see e.g. this FORBES article pointing to industry analysis)

We have added levelized costs of electricity production for coal, gas, solar PV, and wind onshore in the supplementary material in figure S6. Unfortunately it is not clear which article the reviewer is referring to as there was no link to the specific article. We compare to current prices as given by

Lazard.

Likewise, the impact of technological learning rate assumptions on the modeled results is hardly mentioned. The paper would really benefit from a more analytical treatment of the problem presented. Technological learning would also provide easy linkage to the 2 stage decision framework later in the paper. Clearly, the shape of MACs, learning rates and discount rates and the resulting carbon prices interact dynamically calling for a simple analytical model to present the problem before entering exogenous carbon prices into a complicated model.

We agree with the reviewer that it would be valuable to complement our results with an analytical model. However, in our case we are mainly interested in the impact of the carbon price pathway on CDR deployment and whether a less risky pathway can be chosen that approximates the efficient solution regarding both emissions and economic impacts. For this project, features such as different CDR options at different costs (which may increase with deployment, e.g. in case of BECCS where the supply of more bioenergy comes at increasing costs) and with different limitations are important. Path dependencies and a somewhat realistic, slow scale-up of CDR are crucial. These real world factors are not easily represented in a model based on MACs and in an analytical model in general. This would be a full new project that would certainly be interesting to pursue, but that we consider outside the scope of this paper.

The consideration of endogenous technological learning does not affect the Hotelling price path, if the overshoot flexibility is allowed. It only shifts the price path, but does not alter the exponential growth rate. The REMIND model represents endogenous technological learning by default. In an analytical model the consideration of marginal abatement cost curves, endogenous technological learning and CDR in the context of a carbon budget studying the issue of overshoot flexibility implies complex interaction effects and discontinuities in control variables that are difficult to treat in an analytical model. The discontinuity in the control variable is implied, if the carbon budget cannot be overshoot and net emission are kept at zero by the use of sufficient amounts of CDR. The time at which net zero emissions are reached and at what carbon price would depend on the various model parameters, which makes the analytical treatment a challenging task. In a numerical model we include these effects in the scenarios, but we do not have to treat them explicitly.

I did not dig deep enough into the REMIND model and I do not expect the average reader of a NatureComms article to do so. Therefore, it would be interesting to present, in addition, to the consumption loss a few macro-economic indicators such as interest rate, CPI, the share of government expenditures (see minor comment below), the share of industrial production to better comprehend the 3 scenario situations. The main issue I am after here is that as a reader of the paper I get seemingly convinced (mainly because of a clever choice of indicators) that a suboptimal carbon price trajectory is the best choice. The paper should elaborate explicitly and quantitatively on the issue of economic suboptimality and trade-offs of the linear price path e.g. welfare loss.

We have added the optimal carbon price trajectory, which can now be compared directly to the alternatives. We hope that this already mediates the reviewers concern. We also added the interest rate and two energy price indices as additional indicators in the SI.

The decision process presented in the discussion section seems rather decoupled and it seems a little counterintuitive that "The two decisions of immediate carbon pricing and the price increase after 2050 are independent decisions". Technically, this calls for a discontinuity of the carbon price in 2050. There are decision proposals already in the literature mostly stemming from a 2stage stochastic optimization calculus coming to rather different and more elaborate conclusions. I would strongly recommend that the authors familiarize themselves with this literature and embed their

proposal in a setting of 2 stage decision making. Almost surely this will necessitate the authors to propose their own 2 stage decision model which then will be more directly connected to the REMIND model. I am really struggling to see the connection with the numerical results presented above and I am tempted to recommend dropping the discussion section entirely. In fact, I recommend that the 2 stage decision model should be developed with more care and thought for a separate paper.

We agree with the reviewer that the formulation is misleading. Our main interest is in the two independent decisions about the starting price and the shape of the carbon price curve, i.e. the long-term development, though we also added carbon price pathways that have a discontinuity at the point in time when carbon neutrality is reached. The decisions are independent, but at least in the scenario framework taken at the same time. We have reformulated this part and do not call it a 2 stage decision problem anymore to avoid confusion.

Minor issues:

If I evaluate the expenditures necessary to finance 20 GtCO₂ in 2100 (\$4) at a price of 1500 (Fig2) I get 30 Trillion dollars which make ~35% of today's Global Economic Output. How, realistic is such a scenario even if global GDP will be 5-10 fold in 2100?

We agree with the reviewer that this is very challenging and could be considered unrealistic. That is one of the points we are trying to make in the paper, where we are looking for scenarios that reach lower CO₂ prices and lower CDR deployment in 2100. We discuss these finance needs in figure 2. However, in order to avoid cheap showmanship with unrealistically high numbers, we assumed that in such a world with high carbon prices and large amounts of net-negative emissions, a point would be reached where the actual costs of CDR are much lower than the marginal costs of emission mitigation. We therefore assumed as a default that CDR revenues are much lower than the CO₂ price and are cut at 200\$/tCO₂, but we also provide the numbers with the full carbon price in the figure caption. The differentiation of the carbon price for gross emissions and the remuneration for carbon removals can be introduced in a comprehensive policy framework that uses efficient contracts with CDR companies. From a governance perspective this is reasonable because the relatively cheap CDR should only be remunerated at its costs rather than the overall carbon price to avoid huge rent incomes.

The figures in the main text are of poor quality. In some cases, the units on the scales are barely readable.

We have improved the figure quality.

Figure S3 should not be plotted on a logarithmic scale to ensure better and direct comparability with Fig1

The logarithmic scale was chosen to show that the price increase is indeed exponential. The wide spread of carbon prices would make a linear figure barely readable. To improve the direct comparability we have added the two exponential scenarios to the figure.

There are still some typos in the text.

We have carefully revised and edited the text to avoid this.

Figure captions in the SI are too short and lack explanation

We have expanded the figure captions in the SI.

REVIEWER COMMENTS

Reviewer #1 (Remarks to the Author):

Thank you for carefully addressing the issues raised by the reviewers. The new OPT scenario very useful, and I just have few open remarks.

The new optimal OPT scenario is interesting. But lacks some information: given it is optimal it should allow for some negative emissions/overshoot? The carbon price trajectory exhibiting a negative trend after hitting zero makes its interpretation as optimal a bit challenging, unless some hard constraint after hitting were added that alter the dynamics, in which case it can make sense (but should be explained). You write “First, we consider the economically optimal price path evolving endogenously from the model under the condition that cumulative emissions at no point in time exceed the carbon budget. This cost-optimal carbon price path to achieve temperature stabilization shows first an exponential increase until the budget is exhausted. The rate of carbon price increase equals the discount rate.” I would appreciate some explanation here. (1) Adding explicitly prohibiting net negative emissions. (2) The rate of the carbon price explicitly or implicitly equals the discount rate? And need to add “until exhaustion of the carbon budget, since after it even decrease.

- Figure S3: the lines of this study are hardly readable even on screen and should have a darker colour or so.

- I would drop the word “simpler” before alternatives in the abstract as they are alternatives, no necessarily simpler.

- The Golosov et al. (2014) reference (line 73) not sure is correctly references here: They find a growth rate equal to GDP growth due to their simplistic assumptions of constant savings, logarithmic utility, and zero utility discounting, hence just in this particular case equalling gdp growth while actually being the Ramsey discount rate.

- Line 219: “It seems likely that at very high carbon prices, governments would not reward CDR with the full carbon price but try to reduce costs by using an auctioning system.” This is a useful and policy-relevant point. But the auctioning system here would add one more sentence on how its resulting price would differ from the marginal abatement cost.

Reviewer #2 (Remarks to the Author):

I have now, finally after some delay, had time to look at the responses from the authors more closely. Overall, I am now satisfied with the improvements that were implemented by the authors. The paper reads very well, the methods are sound and I have checked the results by inspecting the numbers. I have also inspected the model code that will be published with the paper. However, I was not able to find the data in order to run the model. This might be an oversight from my side. However, I need to emphasis that if the data will not be provided by the authors the paper will not be reproducible. I know that Nature Comm allows for such practices, however, I would consider such practice as a clear knockout criterion for publication. This is an issues for the editors to decide and I do not see myself in the position

make this a requirement. Reproducibility should by now be a standard for publication.

I would still like to provide a few suggestions that could potentially help further improve the paper. I would consider these optional as the paper as such is according to my present judgment good enough to be published.

1) Essentially, the authors say - without directly addressing it - that the carbon budget was wrongly used in the IAM scenarios up to now and argue in a rather convoluted way that the Hotelling rule is not adequate. The text improved a lot already, but the authors could be more direct and precise in their argumentation. I understand that they try to be diplomatic as it is a rather serious issue.

2) The GDP scenario I find interesting, however, there is not enough information on how these actually come about.

3) The Lezard reference for the levelized costs is OK and I think benchmarking the model estimates with these calculations provides a good validation of some of the most crucial assumptions.

4) In the response I read: "We therefore assumed as a default that CDR revenues are much lower than the CO₂ price and are cut at 200\$/tCO₂...". This will need some substantiation as this is a crucial and rather heroic assumption.

Reviewer #1 (Remarks to the Author):

Thank you for carefully addressing the issues raised by the reviewers. The new OPT scenario very useful, and I just have few open remarks.

The new optimal OPT scenario is interesting. But lacks some information: given it is optimal it should allow for some negative emissions/overshoot? The carbon price trajectory exhibiting a negative trend after hitting zero makes its interpretation as optimal a bit challenging, unless some hard constraint after hitting were added that alter the dynamics, in which case it can make sense (but should be explained). You write "First, we consider the economically optimal price path evolving endogenously from the model under the condition that cumulative emissions at no point in time exceed the carbon budget. This cost-optimal carbon price path to achieve temperature stabilization shows first an exponential increase until the budget is exhausted. The rate of carbon price increase equals the discount rate." I would appreciate some explanation here. (1) Adding explicitly prohibiting net negative emissions. (2) The rate of the carbon price explicitly or implicitly equals the discount rate? And need to add "until exhaustion of the carbon budget, since after it even decrease.

We thank the reviewer for their useful comments and the positive evaluation.

To avoid confusion, we have deleted this short explanation in the scenario description and moved it to the results part, where the description of the OPT scenario was already a bit more detailed.

We have expanded the description there, explaining that in the OPT scenario net-negative emissions are not necessary for target compliance and are therefore not realized. They are not prohibited explicitly, but as they are not necessary, the dip in the carbon price trajectory that leads to a stabilization of emissions is economically optimal.

The carbon price evolves endogenously in the model, so it implicitly equals the discount rate. We have clarified in the text that this is the case only until exhaustion of the carbon budget.

- Figure S3: the lines of this study are hardly readable even on screen and should have a darker colour or so.

Thanks for pointing this out. We have changed the color of the individual SR1.5 scenarios to a lighter grey.

- I would drop the word "simpler" before alternatives in the abstract as they are alternatives, no necessarily simpler.

We have deleted the word "simpler" as suggested.

- The Golosov et al. (2014) reference (line 73) not sure is correctly references here: They find a growth rate equal to GDP growth due to their simplistic assumptions of constant savings, logarithmic utility, and zero utility discounting, hence just in this particular case equalling gdp growth while actually being the Ramsey discount rate.

Thank you for pointing out these important assumptions, we have added the assumptions of constant savings and logarithmic utility to the text. It is not clear to us what is meant by "zero utility discounting" as Golosov et al. explicitly show their results in dependence of the discount rate and utility is discounted in the specifications in 2.1.

- Line 219: "It seems likely that at very high carbon prices, governments would not reward CDR with the full carbon price but try to reduce costs by using an auctioning system." This is a useful and policy-relevant point. But the auctioning system here would add one more sentence on how its resulting price would differ from the marginal abatement cost.

We have expanded the text on the auctioning system to explain better how it works and how it would change CDR revenues.

Reviewer #2 (Remarks to the Author):

I have now, finally after some delay, had time to look at the responses from the authors more closely. Overall, I am now satisfied with the improvements that were implemented by the authors. The paper reads very well, the methods are sound and I have checked the results by inspecting the numbers. I have also inspected the model code that will be published with the paper. However, I was not able to find the data in order to run the model. This might be an oversight from my side. However, I need to emphasize that if the data will not be provided by the authors the paper will not be reproducible. I know that Nature Comm allows for such practices, however, I would consider such practice as a clear knockout criterion for publication. This is an issue for the editors to decide and I do not see myself in the position to make this a requirement. Reproducibility should by now be a standard for publication.

I would still like to provide a few suggestions that could potentially help further improve the paper. I would consider these optional as the paper as such is according to my present judgment good enough to be published.

We thank the reviewer for their useful comments and the positive evaluation.

The data has now been published on Zenodo. It contains all model outputs and the scripts to reproduce all figures as stated in the data availability statement. The model code is open source, but some of the data necessary to run the model is proprietary. We are therefore not allowed to make the proprietary input data publicly available at this stage. This is a problem all IAMs are struggling with right now. On a confidential basis, we have provided all necessary files for reviewers to reproduce the model runs here: <https://cloud.pik-potsdam.de/index.php/s/iCpptw75n6AJdSb> Password: CarbonPrice

1) Essentially, the authors say - without directly addressing it - that the carbon budget was wrongly used in the IAM scenarios up to now and argue in a rather convoluted way that the Hotelling rule is not adequate. The text improved a lot already, but the authors could be more direct and precise in their argumentation. I understand that they try to be diplomatic as it is a rather serious issue.

What we are arguing here is, that the Hotelling rule in combination with large-scale CDR leads to a peak-and-decline temperature pathway, which is not well suited for temperature stabilization scenarios. The Hotelling rule has therefore not necessarily been wrong in previous scenarios; it depends on the climate target that was to be achieved. Yet as we know more and more about the difficulties related to such a temperature pathway and the associated large amounts of net-negative emissions, we think that the chosen shape of the carbon price pathway should be carefully considered in future scenarios.

2) The GDP scenario I find interesting, however, there is not enough information on how these actually come about.

In the GDP scenario, the carbon price increases at the rate of GDP growth. We have included this scenario as Golosov et al. and Dietz and Venmans both found this trajectory to be optimal under different assumptions. Though we are not doing a cost-benefit-analysis, we still wanted to add the GDP and the linear scenario that are derived from CBA for comparison.

3) The Lezard reference for the levelized costs is OK and I think benchmarking the model estimates

with these calculations provides a good validation of some of the most crucial assumptions.

Thank you for this assessment.

4) In the response I read: "We therefore assumed as a default that CDR revenues are much lower than the CO₂ price and are cut at 200\$/tCO₂...". This will need some substantiation as this is a crucial and rather heroic assumption.

We have elaborated on the explanation regarding the auctioning system and the resulting finance needs. We are now reporting both numbers, the costs using the current CO₂ price and the costs resulting from a maximum reward for CDR of 250\$/tCO₂ which is a bit higher than our best estimate of long-term costs for direct air capture (~200\$/tCO₂, consisting of 100\$/tCO₂ capital investment costs according to Climeworks and ~100\$/tCO₂ energy costs). This is not the default anymore, but an addition we feel is useful and necessary as a CDR revenue of more than 1000\$/tCO₂ seems unlikely to materialize if actual costs were much lower. The resulting substantial rents for firms deploying CDR would need to be financed by public budgets. This is an assumption about the climate policy framework that is implemented, rather than a techno-economic implication. We show these numbers to avoid exaggerating the finance needs for CDR from public funds by reporting only the finance needs resulting from the actual CO₂ price and therefore show a range of possible outcomes instead. The difference between the case in which carbon removals are rewarded at the same price with which emissions penalized and the case in which they are rewarded at a maximum of 250\$/tCO₂ show that this is indeed a crucial assumption for future climate policy frameworks.

REVIEWERS' COMMENTS

Reviewer #1 (Remarks to the Author):

Thank you very much for the improved discussion on OPT, and explanation. I have no further comments and happy to recommend this paper to be accepted as is for publication.

Reviewer #2 (Remarks to the Author):

Dear Editor,

I have now read the responses to my review comments and I have checked the main manuscript. All issues have been resolved to my satisfaction and I have no further questions.

Kind regards

Reviewer #1 (Remarks to the Author):

Thank you very much for the improved discussion on OPT, and explanation. I have no further comments and happy to recommend this paper to be accepted as is for publication.

Reviewer #2 (Remarks to the Author):

Dear Editor,

I have now read the responses to my review comments and I have checked the main manuscript. All issues have been resolved to my satisfaction and I have no further questions.

Kind regards

We thank the reviewers for their very thorough and helpful comments that have helped to improve the paper substantially.